# Non-Canonical Functions of the Gamma-Tubulin Meshwork in the Regulation of the Nuclear Architecture

**DOI:** 10.3390/cancers12113102

**Published:** 2020-10-23

**Authors:** Matthieu Corvaisier, Maria Alvarado-Kristensson

**Affiliations:** Molecular Pathology, Department of Translational Medicine, Lund University, Skåne University Hospital Malmö, SE-20502 Malmö, Sweden; Matthieu.Corvaisier@med.lu.se

**Keywords:** γtubulin, nuclear architecture, cytoskeleton, nuclearskeleton, cancer, differentiation

## Abstract

**Simple Summary:**

The appearance of a cell is connected to its function. For example, the fusiform of smooth muscle cells is adapted to facilitate muscle contraction, the lobed nucleus in white blood cells assists with the migratory behavior of these immune cells, and the condensed nucleus in sperm aids in their swimming efficiency. Thus, changes in appearance have been used for decades by doctors as a diagnostic method for human cancers. Here, we summarize our knowledge of how a cell maintains the shape of the nuclear compartment. Specifically, we discuss the role of a novel protein meshwork, the gamma-tubulin meshwork, in the regulation of nuclear morphology and as a therapeutic target against cancer.

**Abstract:**

The nuclear architecture describes the organization of the various compartments in the nucleus of eukaryotic cells, where a plethora of processes such as nucleocytoplasmic transport, gene expression, and assembly of ribosomal subunits occur in a dynamic manner. During the different phases of the cell cycle, in post-mitotic cells and after oncogenic transformation, rearrangements of the nuclear architecture take place, and, among other things, these alterations result in reorganization of the chromatin and changes in gene expression. A member of the tubulin family, γtubulin, was first identified as part of a multiprotein complex that allows nucleation of microtubules. However, more than a decade ago, γtubulin was also characterized as a nuclear protein that modulates several crucial processes that affect the architecture of the nucleus. This review presents the latest knowledge regarding changes that arise in the nuclear architecture of healthy cells and under pathological conditions and, more specifically, considers the particular involvement of γtubulin in the modulation of the biology of the nuclear compartment.

## 1. Introduction

The constituents of an organism are determined by the inherited DeoxyriboNucleic Acid (DNA), which is passed from parent to offspring. While the chromosomal DNA in prokaryotes is membraneless, in eukaryotes, the DNA is divided up into several chromosomes and separated from the rest of the cell by a double bilayer of phospholipids, forming two distinct DNA-containing compartments: the nucleus and the mitochondria [1,2]. The mitochondrial (mt)DNA are small circular chromosomes that are organized into nucleoprotein structures and encode for genes that are essential for normal mitochondrial function [3]. This implies that the vast majority of the approximately 3 m long human DNA (3.2 Gb [4]) is compacted in the nucleus in an orderly manner by histone and non-histone proteins that fold the flexible DNA molecule into a chromatin fiber. Additional chemical modifications of the histone proteins cause higher-order compaction of the chromatin [5] and assist in keeping DNA accessible for the maintenance of a controlled gene expression.

For the rapid transmission of environmental signals into a tissue response, spatial organization of the chromatin and gene expression are affected by environmental changes. To this end, in the cytoplasm of a metazoan cell, cytoskeletal elements, such as actin polymers, intermediate filaments, and MicroTubules (MTs), are interlinked and anchored to networks of filaments in the nucleus (nucleoskeleton) such as lamin intermediate filaments [6,7,8,9]. The nucleoskeleton together with the chromatin constitutes the nuclear architecture, which is crucial for maintaining the different appearances of the nuclear compartments of differentiated cells such as granulocytes, megakaryocytes, and fibroblasts [10]. Modifications of the nuclear architecture occur as part of physiological processes such as cell differentiation [10], as well as due to a wide range of pathologies such as cancer [11] and neurodegenerative diseases [12,13]. Knowledge on changes in nuclear morphology has been used for decades by pathologists as a diagnostic tool for various malignancies.

Within eukaryotic cells, the self-polymerizing ability of the protein gamma (γ) tubulin results in the formation of an interlinked protein meshwork, composed of γstrings, γtubules, and centrosomes, in both the cytosol (including all cellular organelles) and the nuclear compartment [14,15,16,17,18]. Here, we review and discuss the latest knowledge regarding changes that arise in the nuclear architecture of healthy cells and under pathological conditions, and, more specifically, consider the particular involvement of γtubulin in the modulation of the biology of the nuclear compartment.

## 2. Nuclear Morphology

The degree of chromatin compaction together with the cytoskeletal/nucleoskeletal architecture mold the nuclear morphology. The different nuclear shapes in the various cell types impact cellular function, and changes in the nuclear shape of a cell type may be linked to various pathologies.

### 2.1. Nuclear Envelope

The delimitating double bilayer of phospholipids enclosing the nuclear chromatin is referred to as the Nuclear Envelope (NE). The Outer Nuclear Membrane (ONM) of the NE is in contact with the cytoplasm and is fused with the endoplasmic reticulum, whereas the Inner Nuclear Membrane (INM) is in contact with the nucleoplasm [19]. The ONM and the INM are separated by a lumen. To ensure bidirectional traffic between the cytoplasm and the nucleus, there are large protein complexes embedded in the NE that form Nuclear Pore Complexes (NPCs), which allow the diffusion of small nonpolar molecules (below the 30–60 kDa size threshold) [20].

Each NPC is formed by multiples copies of proteins called NUcleoPorins (Nups), which are organized to form an ~50 nm open wide channel in the NE [20]. Molecules with a molecular weight of ~35 kDa are able to freely diffuse through the NPC [21], whereas larger molecules require an active mechanism of transport mediated by Nuclear Transport Receptors (NTRs), such as the karyiopherin family [22]. Karyiopherin proteins, such as Importin α, β, or Exportin/ChRomosomal Maintenance 1 (CRM1), recognize and bind to specific amino acid sequences localized on the protein (cargo), which determines its subcellular localization, either in the cytoplasm or the nucleus. Two different amino acid motifs target the carrying protein to a karyiopherin-dependent transport. The first motif is the Nuclear Localization Signal (NLS), which targets the cargo to the nucleus, and the second is the Nuclear Exclusion Signal (NES), which sends the cargo to the cytoplasm. The small GTPase RAs related Nuclear (Ran) regulates the transport of karyiopherins and their cargo proteins across the NPC. This mechanism depends on the nature of the nucleotide bound to Ran, either Guanosine DiphosPhate (GDP) or Guanosine TriPhosphate (GTP). Ran interacts with regulatory proteins such as RAN Binding Protein 1 (RANBP1) and RAN GTPase-Activating Protein 1 (RANAP1) to increase its catalytic activity, and it interacts with Ran guanine nucleotide exchange factor/Regulator of Chromosome Condensation 1 (RCC1) to exchange the hydrolyzed GDP for GTP. The subcellular localizations of these regulatory proteins (RANBP1 and RANGAP1 in the cytoplasm, RCC1 in the nucleus) create a RanGTP gradient, with a high concentration of RanGTP in the nucleus and a high concentration of RanGDP in the cytoplasm [23]. Importins bind to their cargo under the low concentration of RanGTP in the cytoplasm and release it in the nucleus where the concentration of RanGTP is higher. Exportins work in the opposite manner, releasing their cargo in the cytoplasm.

Inside the nucleus, the INM is in direct contact with both the chromatin and the intermediate nuclearskeletal filaments, named the nuclear lamina. In mammalian cells, the nuclear lamina is a structural meshwork composed of lamin A, B1, B2, and C. Lamin A and C are alternative splice variants of the *LMNA* gene, whereas lamin B1 and B2 are encoded by *LMNB1* and *LMNB2* genes, respectively [24]. Changes in the nuclear shape occur following mutations of *LMNA* [25,26] or knockdown of lamin B1 [27]. Lamin B provides elastic properties, supporting NE deformation, whereas LaminA and C determine the stiffness of the cell nucleus [28,29].

### 2.2. Force Balance between Cytoskeleton and Nucleoskeleton

Outside the nucleus, cytoplasmic filaments are connected to the nuclear lamina through the LInker of Nucleo- and Cytoskeleton complexes (LINCs), which act as protein bridges that connect the cytoskeleton with the nucleoskeleton [30]. LINCs are composed of two families of interacting transmembrane proteins: Sad1 UNc-84 domain protein (SUN) proteins embedded in the INM and KASH (named Nesprin in mammal cells) proteins embedded in the ONM. While SUN proteins interact with the nuclear lamina and NPCs [31,32], KASH proteins interact with all cytoskeletal components, including actin filaments [7], intermediate filaments [8], MTs [9], and the γtubulin meshwork [33]. These interactions allow actin stress fibers to exert both contractile and compressive forces, whereas the MTs employ compressive forces on the nucleus, thereby affecting the plasticity, size, shape, and chromatin organization of this cellular compartment [34].

The force balance created by the link between lamin proteins and the cytoskeleton through the LINCs is altered in various laminopathies such as Hutchinson–Gilford Progeria Syndrome (HGPS) [35,36]. HGPS is the consequence of the accumulation of a lamina A mutant isoform termed progerin, which results in an abnormal nuclear morphology driven by the MT network [36]. In *Drosophila* embryos, polymerization of cytosolic MTs in bundles caused deformations of the NE and affects the dynamics of chromatin [37]. Thus, the cytoplasm, the LINCs, the NE, the NPCs, and the nuclear lamina are tightly associated, and modulation of one of these components alters the force balance, causing changes in the nuclear architecture.

### 2.3. Nuclear Bodies

Nuclear chromatin is further compartmentalized in membraneless nuclear domains known as the Nuclear Bodies (NBs) [38]. The absence of lipid boundaries delineating an NB supports the view that the appearance and the structural maintenance of the NBs is the result of self-association properties of the components in the NB. NBs, such as nucleolus, Cajal bodies, and ProMyelocytic Leukemia protein (PML) bodies, are specialized to perform specific nuclear processes, and their structures are maintained by protein–RiboNucleic Acid (RNA) interactions. The size, shape, and number of NB’s varies depending on the cell type and tissue and may change in response to cellular conditions. For example, nucleoli are formed at the end of mitosis around Ribosomal (r) DNA repeats, named Nucleolus Organizer Regions (NOR), on multiple chromosomes that cluster in response to transcriptional activity of these genes [39,40]. The nucleolus harbors a tripartite architecture with three distinct areas: the inner part of the nucleolus is called the Fibrillar Centre (FC), the outer is the Granular Centre (GC), and in-between them is the Dense Fibrillar Component (DFC) [41]. Loss of the integral nuclear envelope protein SUN1 alters this structure [42,43]. The main function of a nucleolus is to assemble the transcription and processing machineries that are responsible for generating ribosome subunits. Transcription of rRNA mediated by RNA Polymerase I occurs at the interface of the FC and DFC, whereas maturation and association of rRNA with proteins to form premature ribosomal subunits occur within both the GC and DFC [41]. The number of nucleoli changes in different tissues and cell types, as well as during cell proliferation [44,45].

## 3. Show Me Your Nuclear Architecture and I Will Tell You Who You Are

The dynamic interactions between cytoskeletons and the NE with the NPC inside it and the lamina below in connection with the cytoskeleton through LINCs as well as the chromatin and chromatin-associated RNA and proteins included in NBs influence the nuclear architecture depending on the activity of a cell and its environment. This complex balance leads to the varying appearance of the nuclear compartment in different cell types. For example, the fusiform nucleus of smooth muscle is adapted to facilitate muscle contraction. Other examples are the lobed nucleus in leukocytes, which assists with the migratory behavior of these immune cells, and the condensed nucleus in sperm, which aids in their swimming efficiency [10]. The nuclear architecture of cells is also altered during disease development, and the aberrant changes in the nuclear morphology have been used for decades as a diagnostic tool. One such tool is the PAPanicolaou (PAP) smear test, which is based on aberrant changes in the morphology of neoplastic cells and is used by pathologists to diagnosed cervical cancer [46].

### 3.1. The Organization of the Nucleus Influences Gene Expression

In situ hybridization and chromosome painting techniques have revealed that the nuclear chromatin is organized in chromosome territories that occupy well defined nuclear regions, establishing spatial patterns [47]. There is a tendency for small or gene-rich chromosomes to be located towards the interior, whereas larger or gene-poor chromosomes are positioned next to the nuclear periphery [48]. This organization results in the positioning of genes in the nuclear interior, for example, NBs are found in the interchromatin region of the nuclei, whereas tightly packed chromatin, known as heterochromatin, which plays an important role in nuclear architecture and gene silencing, is either randomly distributed or localized near the nuclear periphery [49,50,51]. This spatial pattern is partially kept by the lamina, as demonstrated by an analysis of *LMNA* mutant mice that had lost the expression of Lamin A and Lamin C. In these mice, cells had misshaped nuclei, and the repartition of heterochromatin foci next to the INM was also altered [25,26]. Work in human fibroblasts [52] and drosophila melanogaster cells [53] confirmed that lamins interact with specific Lamin-associated DNA domains (LAD) [54], which are regions with a low density of genes, as reflected by the presence of low amounts of two markers for active transcription: RNA polymerase II, and methylated histone 3 at lysine 4 [52,53]. The anchoring of chromosomes with the NE provides a supporting platform for the chromosomes. Chromosome–NE interactions affect the degree of chromatin movements, resulting in a plastic cell nucleus with more dynamic chromatin in stem cells, and stiffer and lineage-specific chromosome arrangements in differentiated cells [55,56,57].

Next to INM and the lamina meshwork, around NPCs there exists a gene-transcription-prone environment, known as euchromatin, which is partially maintained through the functions of one of the components of the NPC Nuclear Basket (NB): the protein Translocated Promoter Region (TPR) [58]. This microenvironment allows, for example, the recruitment of the transcription factor avian MYeloCytomatosis viral oncogene homolog (MYC) to the NB, which ensures the formation of active transcription complexes that facilitate both proliferation and migration [59]. In this way, genes involved in specific transcription pathways are spatially close to each other and to the transcriptional and post-translational machinery [60]. During oncogenic-induced senescence, TPR mediates the formation of the Senescence Associated Heterochromatin Foci (SAHF), which delocalizes peripheral heterochromatin to inner parts of the nucleus [61].

Alterations in the organization of chromosomes and of the nuclear morphology are associated with tumor progression. Specific chromosomal translocations increase the amount of heterochromatin and alter the nuclear lamina, contributing to the formation of misshaped nuclei [62]. The new genomic rearrangements can create proto-oncogenic gene clusters that are transcriptionally active [63]. Additionally, the chromatin organization of nuclear bodies is altered, and NBs become larger and numerous [11].

### 3.2. During Differentiation

Environmental, mechanical, chemical, and biological cues trigger cell differentiation. These extracellular signals are transmitted by the cytoskeleton into the nuclear compartment, resulting in changes in gene expression. Cytoskeleton networks provide the mechanical support for the cytosolic transmission of the signal but can also transmit force, which affects the nuclear architecture through LINCs. Major changes in the cytoskeleton occur during the transition from stem cells to differentiated offprints, and these changes may facilitate differentiation.

Actin and MTs are highly dynamic cytoskeletons that can rapidly change after a signal is initiated. Indeed, during mesenchymal stem cell differentiation, increased numbers of cytosolic stress fibers formed from actin promote osteoblastic differentiation, whereas disruption of actin polymerization with small molecule inhibitors or actin–myosin (a motor protein) interactions favored adipogenic differentiation [64,65,66]. Actin is also present in the nucleus, where it regulates nuclear processes such as transcriptional regulation and chromatin remodeling, leading to the differentiation and development of mesenchymal stem cells and epidermal progenitors [67].

Cellular differentiation also causes the reorganization of MTs. In eukaryotic cells, an MT usually consists of 13 laterally associated protofilaments that form hollow tubes in the cytoplasm, axons, and mitotic spindles [10]. Each protofilament is made up of α- and β-tubulin heterodimers. MTs are polar structures, and a large number of MT-associated proteins regulate the dynamic behavior of the plus and minus ends [68]. In cells, a new MT nucleates from the minus end on a complex formed of γtubulin and various γtubulin complex proteins (GCPs); this is known as the γtubulin ring complex (γTURC). The γTURC–MT interactions can remain after the MT is formed and can both cap the minus ends of non-centrosomal MTs or anchor MTs to MicroTubule-Organizing Centers (MTOC; centrosomes in animal cells and spindle pole bodies in fungi) [69,70]. γTURCs are enriched at the centrosome, and these membraneless organelles are specialized to ensure a higher rate of MT polymerization in proliferative cells [71]. However, differentiation reorganizes MTs from centrosomal into non-centrosomal MTs arrays, and this altered organization causes, for example, during granulopoiesis, the interaction between MTs and the NE to influence the nuclear shape by mediating nuclear lobulation [72].

Alterations in the nuclear architecture may result in differentiation into a specific lineage or in the development of pathologies due to, for instance, modification of lamin functions or an altered nucleocytoplasmic transport. Large scale mapping of the interaction of nuclear lamina with chromatin demonstrates the dynamic changes in the localization and geography of the chromatin in the nucleus during different stages of differentiation [73,74]. Changes in the expression of nucleoplasmic proteins in NPCs, such as Nup153, contribute to the maintenance of pluripotency by repressing the expression of genes necessary for cellular differentiation in embryonic stem cells, independently of the cargo [75]. Nup153 interacts with transcription factors such as Sox2 [76] or with proteins such as the Polycomb Repressive Complexes (PRC1) [75], and depletion of Nup153 increases the expression of genes necessary for the differentiation of Neural Stem Cells into neuronal and glial cells [76]. In this way, Nup153 modulates gene expression and thereby forms a pluripotency/differentiation switch that affects the architecture of the chromatin.

The complex balance between cell proliferation and differentiation drives commitment into a lineage while ending cell proliferation. Thus, impeded lineage differentiation may result in cell proliferation and cancer, and these changes are reflected in the morphology and architecture of cells [77,78]. The degree of differentiation in a tumor is used to distinguish malignant from benign tumors, and it is a central aspect of the histopathological classification of solid tumors [79].

## 4. The Gamma-Tubulin Meshwork

The compartmentalization of eukaryotic cells into several organelles creates the need for structure, communication, and transport between compartments. In the cytoplasm and nucleus, three main families of structures ensure these functions—actin filaments, intermediate filaments, and MTs—and the LINCs are one of the linking structures between compartments [80]. In addition, the γtubulin meshwork establishes a connection between the cytoplasmic and nuclear compartments [14,17,18].

### 4.1. Tubulins

γTubulin is part of a family of GTPases called the tubulins, which are involved in shaping the architecture of the human centrosome (Table 1) [81,82]. In humans, there are five known tubulin isoforms, αtubulin, βtubulin, γtubulin, δtubulin, and εtubulin [83], of which only α-, β-, and γ-tubulins are ubiquitous. Multiple genes encode for α- and for β-tubulin, but the number of genes encoding γtubulin ranges from one to three, with two genes found in mammals and up to three genes found in flowering plants (http://genome.ucsc.edu/) [84]. Studies of human U2OS osteosarcoma cells and murine NIH3T3 embryonic fibroblasts revealed that in the tubulin family, γtubulin is the only member that contains an NLS [85] and a helix-loop-helix DNA-binding motif on the C terminus [86]. In humans, two proteins, γtubulin1 and γtubulin2, are encoded by two genes: *TUBG1* and *TUBG2*. γTubulin1 is a ubiquitously expressed protein, whereas γtubulin2 is highly expressed in the brain [87]. γTubulin1 and γtubulin2 proteins exhibit an amino acid level identity of 97%, and the main differences in the protein sequence are localized to the DNA binding domain at the C-terminus of the protein [14,88].

The shared characteristics among αtubulin, βtubulin, and γtubulin explain the similarities between MTs and the γtubulin meshwork. In eukaryotic cells, α- and β-tubulin heterodimers and monomers of γtubulins form protofilaments [87,89,90,91] and γstrings, respectively (Figure 1). γStrings are located in the cytoplasm, the PeriCentriolar Material (PCM) of the centrosome, and the nuclear compartment (Figure 1 and Figure 2 and Table 1) [14,17,18], whereas cytosolic α- and β-tubulin protofilaments nucleate on a γTuRC [87,89,90,91] to form a MT (Table 1) [92,93,94]. In mammalian cells, in the absence of α- and β-tubulin heterodimers, γTuRCs and pericentrin assemble a γtubule (Figure 2), resulting in the formation of a fiber of similar size as an MT [16]. γTuRCs are found in the cytoplasm and the centrosomes, and are also associated with cellular membranes [87,89,90,91]. In contrast to γstrings, which are static structures, γtubules and MTs are temperature-sensitive polar structures that vary both in size and location, and both can emanate from centrosomes (Figure 2) [16,95].
cancers-12-03102-t001_Table 1Table 1Proteins and protein complexes that both interact with γtubulin and affect nuclear structure.Protein/ComplexReferenceCommentsCentrosome[82,83]Cytoskeletal organizing centersMicrotubules[92,93,94]Polymers of tubulinsγTuRC[87,89,90,91]MT-nucleating unitγTubule[16]γTubulin-rich filamentsγStrings[17,18,96]Thin cytosolic/nuclear γtubulin threadsActin[97]Cytoskeletal/nuclearskeletal elementIntermediate filaments[98]CytoskeletonLamin B[17]NuclearskeletonRan[99]Regulates transport across NPCMel-28/ELYS[99]Required for NPC assemblyNucleolin[100,101]RNA-binding protein at the nucleolusE2F[88,102,103]Regulates gene expressionSUN[33]Links the nuclear lamina with NPCsSamp1[104,105]Inner nuclear membrane protein

### 4.2. The Dynamics of the Gamma-Tubulin Meshwork

Cellular γtubulin has been described as being associated with all of the following compartments: the nucleus, the Golgi, the endoplasmic reticulum, the endosomes, the mitochondria, and the centrosomes (Table 1) [18,88,89,90,91,101,104,106,107]. Due to its self-polymerizing features, γtubulin produced by bacteria assembles in vitro γstrings that support the formation of lamin B3 protofilaments (Table 1) [17].

During cell division, the inherited centrosome and genome duplicate synchronously in the S-phase. At the onset of mitosis, the two centrosomes ensure the assembly of a bipolar mitotic spindle and the strict segregation of sister chromatids between offspring cells, resulting in two cells with one centrosome and one genome set each. During nuclear formation in mammalian cell lines and *X. laevis* egg extracts, γstrings establish a nuclear protein boundary around chromatin that connects the cytoplasm and the nuclear compartment together throughout interphase (Figure 1 and Table 1) [17]. This chromatin-associated γstring boundary serves as a supporting scaffold for the formation of a NE around chromatin by facilitating the nucleation of lamin B1 during nuclear formation. Moreover, at the NE, γtubulin is associated with Ran and with nuclear pore proteins (Figure 1 and Table 1) [99]. In the nuclear compartment, mass spectrometry analyses of purified fractions of human nucleoli identified γtubulin at that location together with nucleolin, the most abundant RNA-binding protein at that site (Figure 1 and Table 1) [100,101]. At the G1/S transition, the phosphorylation of γtubulin on Ser^131^ and Ser^385^ regulates the recruitment of this protein to the nascent centriole, where it enables centrosome replication and also promotes the accumulation of γtubulin in the nucleus [85,88,100,106,108,109,110,111]. In human U2OS osteosarcoma cells, mutations in the GTP/magnesium-interacting residue Cys^13^ of γtubulin are cytotoxic [16,18,112,113], and treatment with either of the γtubulin GTPase binding domain inhibitors Citral Dimethyl Acetyl (CDA) and DiMethyl Fumarate (DMF,) disassembles γtubules [16,95]. These observations strongly suggest that the GTPase domain of γtubulin is essential for the dynamics of the γtubulin meshwork. The “where” and “when” characteristics of the self-polymerizing ability of γtubulin are most likely regulated by GTP acting together with phosphorylation-dependent changes in the conformation of γtubulin.

### 4.3. The Gamma-Tubulin Meshwork and Gene Transcription

In a genetic screen to identify proteins required for the proper functioning of homeotic genes in *drosophila melanogaster*, mutations in the *brahma* (*brm*) gene, a gene product related to the chromatin remodeling complex SWI/SNF, showed a genetic interaction with *γTub23C* (γtubulin1) mutations [114], suggesting a role of γtubulin1 in transcription. In this context, γtubulin interacts with the transcription factor family E2F in animals and plants (Figure 1 and Table 1) [88,102,103]. Indeed, nuclear γtubulin was found to bind to the DNA on the same DNA binding motif as E2F, leading to the view that the tumor suppressor retinoblastoma (RB)1 and γtubulin proteins complement each other in the regulation of gene expression [108]. The RB1/γtubulin signal network governs E2Fs, whose transcriptional activities induce the expression of the target genes that are indispensable for centrosome duplication and DNA replication [115]. Interestingly, besides interacting with E2Fs, RB1 can recruit remodeling factors, including histone deacetylases, members of the chromatin remodeling complex SWI/SNF, and DNA methyltransferase, aiding in the modification of the structure and organization of chromatin [116,117,118]. Altogether, these data indicate that γtubulin may be involved in the recruitment of DNA-remodeling factors.

In the nucleoli, electron microscopy experiments, performed by Hořejší and colleagues [100], showed that nucleolar γtubulin is present in the GC, where RNA transcription takes place (Table 1). It was also shown that, at that position, γtubulin is localized with the tumor suppressor C53, and this interaction is necessary for modulating the activity of C53 after treatment with DNA-damaging compounds, suggesting that the activity of γtubulin is necessary for DNA repair. In line with this assumption, γtubulin associates with Rad51, BReast CAncer type 1 susceptibility protein (BRCA)1, p53, Checkpoint Kinase (Chk)2, and Ataxia Telangiectasia and Ataxia Telangiectasia and Rad3-related protein (ATR), which are proteins that are involved in checkpoint activation and DNA repair [100,106,119,120,121,122,123,124,125].

### 4.4. The Gamma-Tubulin Meshwork and Nuclear Architecture

Depletion of γtubulin in *Xenopus laevis* egg extracts was shown to impair nuclear membrane formation [17]. Furthermore, live imaging of cells expressing γtubulin1 mutants, showed that impairment of the γstring boundary around chromatin led to the formation of chromatin empty nuclear-like structures, which collapsed into cytosolic lamin aggregates [17]. Moreover, in *A. thaliana*, γtubulin was found to be colocalized with SUN1 in the INM (Figure 1 and Table 1) [33]. It is worth noting that at the INM, both SUN1 and γtubulin interact with Samp1 and the nuclear lamina (Figure 1) [104,105]. Depletion of Samp1 impairs the proper recruitment of γtubulin to the mitotic spindle, revealing an important link between the interaction of nuclear proteins with γtubulin for the completion of cell division (Table 1) [104]. In the NE, γtubulin interacts with Mel-28/ELYS (Figure 1 and Table 1) [99], a Nup protein whose depletion impairs the proper formation of the NPC due to the inefficient recruitment of the Nup107-160 complex [126].

The fact that the PCM and γstrings are important sites that affect the nucleation and dynamics of MTs, actin filaments, and intermediate filaments (Table 1) [17,97,98], suggests that γstrings may function as a nucleating platform. With this in mind, the large number of γstrings associated with cellular membranes may support those membranes and provide a nucleating platform for cytoskeleton elements such as actin, MTs, or lamins, allowing them to mold and transmit signals into different cellular compartments (Figure 1 and Table 1). Accordingly, the interaction of γTURC with the Golgi membrane-linked GMAP-210 protein regulates the proper positioning and biogenesis of the Golgi apparatus [91]. In addition, γtubulin is associated with endosomes [89], and it is an important mitochondrial infrastructure that connects the mtDNA with the nuclear chromatin (Figure 1) [18,90]. Similarly, DNA-bound γstrings fasten the chromatin to the cytosol, and the site for the formation of a NE around the chromatin is marked by the transition between cytosolic and nuclear-associated γstrings [17,99,127,128]. Accordingly, in U2OS cells, treatment with either CDA or DMF (to increase the endogenous levels of the metabolite fumarate [103,129]), disassembled γstrings, disrupting the association between mitochondria and the nuclear compartment [18]. Altogether, these data support the notion that γtubulin is indispensable for structuring cellular compartments and coordinating the cytoplasm with the nuclear compartment.

Finally, in the cytoplasm, various γtubules emanating from a centrosome can intertwine, forming macro-γtubules, and the formation of those structures may influence the shape of the NE [14]. Altogether, findings may suggest that cytosolic γtubules, NE-inserted γstrings, and centrosomes work as structural docking sites for nuclear γtubulin, resulting in the recruitment of chromatin-remodeling factors necessary for the remodeling of the genome (Figure 1 and Figure 2). 

### 4.5. The Gamma-Tubulin Meshwork and Cell Differentiation

Among the *TUBG* genes, *TUBG1* is the most ubiquitous and predominantly expressed gene among all species, whereas *TUBG2* and *TUBG3* are expressed in the human brain and flowering plants, respectively. This implies that the expression of *TUBG2* and *TUBG3* isoforms are restricted to differentiated cells [84,87]. Indeed, in mice, *TUBG1* is expressed in the cortex during embryonic development, but the expression of *TUBG2* increases as the brain develops [130]. Differentiated neuroblastoma cells also express higher levels of γtubulin2, which further supports the idea that γtubulin1 and γtubulin2 have different functions during differentiation [90].

*TUBG1* knock-out in mice is lethal. In contrast, *TUBG2* knock-out mice are viable but exhibit some defects, including abnormalities in their circadian rhythm and painful stimulation [87]. In humans, mutations in the *TUBG1* gene have been reported in children suffering from malformations related to cortical development [131,132]. Mice expressing the *TUBG1* pathogenic variants suffer from cortex malformation and behavioral defects [130], proving the importance of γtubulin proteins in the development of the central nervous system.

A study conducted in isolated hippocampal neurons has shown that during the process of maturation, the centrosomal expression of γtubulin decreased after two weeks in culture [133]. Furthermore, in neurons, the nucleation of MTs does not rely on the centrosomal functions of γtubulin [134,135]. In general, in differentiated cells, there is a loss of MTOC activity in the centrosome and in the novel acquisition of MTOC activity at other cellular sites. This is partially achieved by altering the centrosomal localization of various proteins present in the PCM, which can be achieved in various ways [136,137,138]. For example, during differentiation of the mammalian epidermis, there is a transcriptional downregulation of genes encoding centrosomal proteins [139], whereas in neurons, the alternative splicing of the centrosome-targeting domain of the centrosomal protein ninein results in ninein dispersal [140]. In *Drosophila* oocytes, the MTOC pushes the nucleus, causing the formation of a groove on the NE, which results in the migration of the nucleus and the establishment of a dorsal-ventral axis [141].

### 4.6. Gamma-Tubulin in Cancer

As described above, γtubulin is involved in many crucial cellular processes, including cell proliferation and differentiation, processes that are hijacked during oncogenesis [142]. At the genomic level, few mutations or amplifications of *TUBG* genes have been reported in patients with cancer. According to the cbioportal [143], *TUBG1* and *TUBG2* have been observed to be amplified in rare subtypes of breast cancer (2 cases of adenoid cystic adenocarcinoma, *N* = 16 patients) and prostate cancer (Neuroendocrine Prostate Cancer, 19 cases, *N* = 114). Moreover, high levels of *TUBG1* mRNA coincide with high levels of cell cycle-related genes in various tumor types, supporting the notion that γtubulin1 expression is necessary for proliferation [88]. Accordingly, in various tumors (retinoblastoma, bladder, breast, colorectal and small cell lung carcinoma (SCLC) tumors), γtubulin and RB1 moderate each other’s expression, and in the absence of γtubulin and RB1, the uncontrolled transcriptional activity of E2Fs upregulates apoptotic genes, causing cell death [88,108]. The RB pathway is one of the most well-described tumor suppressor pathways, and it is found to be highly mutated in a large spectrum of cancers [144,145]. Using the cbioportal [143], we found that among 31 subtypes of cancer, 30% exhibit defects in the RB pathway, with 7% of the patients exhibiting a loss of *RB1* [144]. In brain tumors, high expression of γtubulin is associated with high-grade astrocytomas and glioblastomas, as compared to low-grade astrocytomas, which exhibited weaker γtubulin staining [146]. We can also point out that in medulloblastoma samples in areas of the tumors with high staining of the neuronal differentiation marker β3tubulin, the staining of γtubulin is low, whereas in tumor areas with low staining of β3tubulin, there is a strong co-staining of γtubulin and the proliferative marker proliferating cell nuclear antigen (PCNA) further confirming that there is a connection between γtubulin expression and highly proliferative tumors [147].

It is tempting to speculate that the high expression of γtubulin1 in proliferating cells may provide chromatin with more anchoring sites at the NE, which might assist in the maintenance of a plastic cell nucleus in proliferating cells, whereas, low expression of γtubulin1 will result in a stiffer nucleus that favors lineage-specific chromosome arrangements in less aggressive tumor cells [55].

Due to the functions of tubulins in the biology of the cell and, specifically, due to their involvement in mitotic spindle formation, since the 1960s, MTs have been the target of various chemotherapies that impair MT function, cell division, and angiogenesis, and thus reduce tumor growth [148,149,150]. These compounds are widely prescribed as antineoplastic drugs for a broad range of malignancies including lung, breast, gastric, esophageal, bladder, and prostate cancers, Kaposi’s sarcoma, and squamous cell carcinoma of the head and neck [149]. However, the effectiveness of MT-targeting drugs for cancer therapy is limited by drug resistance and severe side effects in treated patients [149]. The target for MT-targeting drugs is α/β tubulins heterodimers, but more recently, MT-targeting drugs that inhibit γtubulin have also been developed [151].

The recently discovered functions of γtubulin in the nucleus and their inverse correlation with the tumor suppressor RB1, together with the high expression of γtubulin in proliferating cells suggest that drugs specifically designed to inhibit the nuclear activity of γtubulin may pave the way for chemotherapies that target a broad range of malignant tumors but have no impact on healthy cells. The natural product, citral, affects αβ- and γtubulin activities [152], but the citral analog, CDA, specifically inhibits the nuclear activities of γtubulin without affecting MT dynamics [103]. CDA has been proven to have an in vivo antitumorigenic activity. Furthermore, the drug dimethyl fumarate (DMF) also targets the nuclear activities of γtubulin. DMF is a Food and Drug Administration (FDA)-approved drug for the treatment of multiple sclerosis and psoriasis [153,154] and, in addition, has been reported to diminish melanoma growth and metastasis in animal models [155]. Thus, based on the functions of γtubulin in the nucleus, the development of drugs that inhibit its nuclear activity may act specifically on tumor cells while sparing healthy tissue.

## 5. Conclusions and Perspectives

Alterations in the nuclear cell morphology are signs of malignancy and are used as a diagnostic method for human cancers. The equilibrium between proliferation and differentiation is reflected in the appearance of the nucleus and is controlled by the balance established among expressed proteins in a cell. For example, the expression of different αtubulin and βtubulin genes in the different tissues provides fine tunes for the functions of MTs [84].

This review summarizes the known functions of γtubulin in the regulation of cell morphology and differentiation. However, our knowledge on the functions of the different γtubulin isoforms in controlling the nuclear architecture and cell differentiation is limited. Consequently, we need to gain more knowledge on the potential roles of the γtubulin meshwork in health and disease, as this may aid in the discovery of novel therapeutic regimens that target the activities of γtubulin.

## Figures and Tables

**Figure 1 cancers-12-03102-f001:**
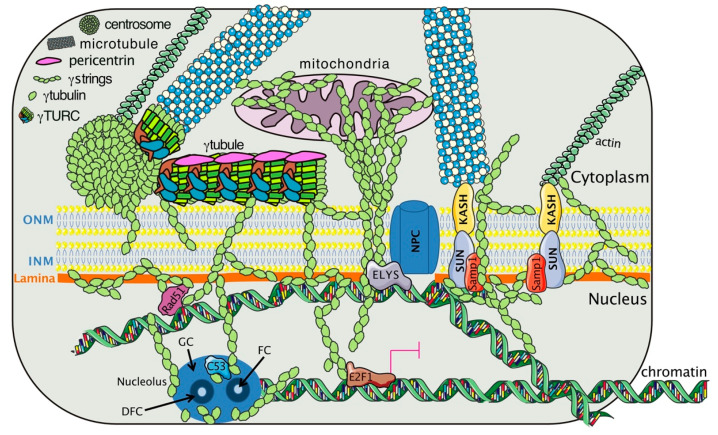
The γtubulin meshwork interacts with cellular components that affect the nuclear architecture. The meshwork is composed of centrosomes, γtubules, and γstrings. A γtubule consists of a γtubulin ring complex (γTURC), and pericentrin and can lie close to the outer nuclear membrane (ONM). Hypothetical representation of how γstrings connect cytosolic organelles (centrosome, mitochondria) and cytoskeletal elements (MTs, actin, and γtubules) with the nuclear compartment [14,17,18]. In the nucleus, γtubulin interacts with laminB (lamina) [17], the nucleoporin embryonic large molecule derived from yolk sac (ELYS), which is part of the nuclear pore complex (NPC) [99], the inner nuclear membrane (INM) protein Samp1 [104], the chromatin associated proteins Rad51 [106] and C53 [100], and the transcription factor E2 promoter binding factor (E2F)1 [88]. The black arrows indicate the positions of the fibrillar center (FC), granular center (GC), and dense fibrillar component (DFC) in the nucleolus. The magenta lines indicate inhibition.

**Figure 2 cancers-12-03102-f002:**
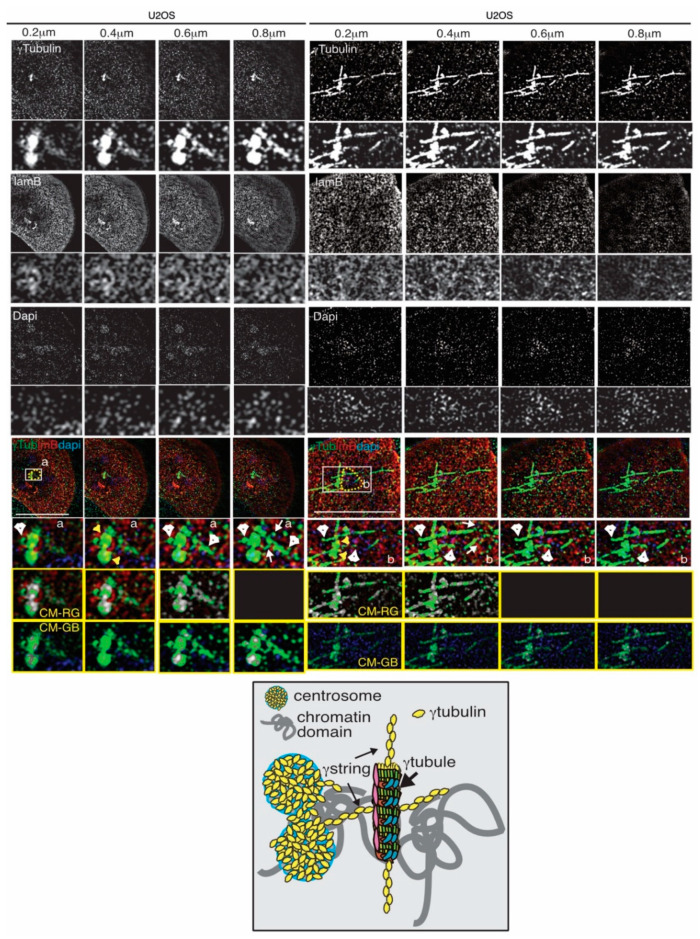
Chromosomes, centrosomes, γtubules, and γstrings are connected and facilitate the formation of chromatin domains (yellow dashed line). Structured illumination microscopy images of two fixed U2OS cells show the localization of endogenous γtubulin with an anti-γtubulin (γTub), lamina with the anti-laminB (lmB; a marker of the nuclear membrane) antibody, and DNA with DAPI in U2OS cells in the interphase. Sequential confocal images (6 stacks per cell) were collected at 0.2 μm intervals to enable a comparison of the network of fibers. White arrowheads and arrows show γtubules and γstrings, respectively, and yellow arrows indicate the centrosomes. White boxes show the magnified areas displayed in the insets. The scale bar is 10 μm. The yellow boxes show colocalized pixel maps (CM) of the red (R), green (G), and blue (B) channels of the magnified areas illustrated in the inset. Grey areas in the maps denote pixels colocalized between channels. The cartoon represents interactions between γtubules, γstrings, and centrosomes at the NE, which act as a protein scaffold for the formation of chromatin domains.

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
