# Peer review of "Non-Canonical Functions of the Gamma-Tubulin Meshwork in the Regulation of the Nuclear Architecture"

_cancers, 2020, doi:10.3390/cancers12113102_

Round 1
Reviewer 1 Report
This review summarizes non-canonical functions of the γ-tubulin meshwork. I enjoyed reading it. The manuscript is interesting and informative, but in order to provide the readership a better overview about the role of γ-tubulin meshwork I would suggest adding a table showing main roles of γ-tubulin .
Some minor comments:
1) Abstract: γ-tubulin instead of tubulin.
2) The use of abbreviations is unsystematic and makes reading difficult. Section "Abrreviations" does not contain all relevant abbreviations.
3) Row 388 - title is incorrect
4) Row 421- something is missing
Author Response
Reviewer point #1
This review summarizes non-canonical functions of the γ-tubulin meshwork. I enjoyed reading it. The manuscript is interesting and informative, but in order to provide the readership a better overview about the role of γ-tubulin meshwork I would suggest adding a table showing main roles of γ-tubulin.
Author response #1
Thank you for your comments. We have included a table summarizing the proteins that interact directly with γtubulin. This together with figure 1 gives a good overview of the cellular functions of gamma-tubulin
Some minor comments:
Reviewer point #2
1) Abstract: γ-tubulin instead of tubulin.
Author response #2
We have noticed that the uploaded files suffered some changes during the submission process, which explain the mentioned mistake. We have mended the problem in the revised version of the manuscript.
Reviewer point #3
2) The use of abbreviations is unsystematic and makes reading difficult. Section "Abrreviations" does not contain all relevant abbreviations.
Author response #3
Thank you for your comment. We have added the missed abbreviations and systematized the used in the text.
Reviewer point #4
3) Row 388 - title is incorrect
Author response #4
We have added the missed information to the title. This dependence once more on the changes that the uploaded files suffered.
Reviewer point #5
4) Row 421- something is missing
Author response #4
We have revised the text.
Thank you for your time and consideration.
Reviewer 2 Report
This review focused on the functions of the nuclear g-tubulin and presents the latest knowledge regarding changes that arise in the nuclear architecture of healthy cells and under pathological conditions. In addition, the authors also mentioned about the particular involvement of tubulin in the modulation of the nuclear compartment. This is an interesting reviews that summarizes in a concise manner the data about noncanonical function of tubulin. This timely subject is covered in a comprehensive and a clear manner.
- The authors mentioned “The mitochondrial (mt)DNA are small circular choromosomes that are organized into nucleoprotein structures” (line28). This sentence requires references.
- The authors mentioned ”in the cytoplasm of a metazoan cells, cytoskeletal elements, such as actin polymers, intermediate filaments, and microtubules, are interlinked and anchored to networks of filaments in the nucleus (nucleoskeleton) (line 36)”. How are actin polymers, intermediate filaments, and microtubules anchored to nucleoskeleton?
- Line 68, there are unexpected spaces, I guess.
- I think “RAs” might be Ras (line 73), and “The small GTPase Ras related nuclear protein (Ran)” is better.
- Line 77, I think “or” should be “and”.
- I think that AP1 shoud be RanGAP1 (line 77).
- Line 88. RanGEF has specific name, RCC1.
- Line 95, there are unexpected spaces, before TURC.
- Figure 1 shows g-tubulin in the space between INM and ONM. Are there any references?
- Line 98, there are unexpected spaces, just before tubulin.
- Line 124-140, the authors mentioned about nucleoli. Regarding to the subject of this review, I think that it had better to mention that the relationship between nucleoli and LINC complex.
- The subtitle “Show me your nuclear architecture and I will tell you who you are” (line 141) is different from other subtitles.
Author Response
Reviewer point #1
The authors mentioned “The mitochondrial (mt)DNA are small circular choromosomes that are organized into nucleoprotein structures” (line28). This sentence requires references.
Author response #1
We have added a reference to the sentence
Reviewer point #2
The authors mentioned ”in the cytoplasm of a metazoan cells, cytoskeletal elements, such as actin polymers, intermediate filaments, and microtubules, are interlinked and anchored to networks of filaments in the nucleus (nucleoskeleton) (line 36)”. How are actin polymers, intermediate filaments, and microtubules anchored to nucleoskeleton?
Author response #2
We have added the references to this sentence. This is part of the introduction and thus we just want to highlight the points that affect nuclear architecture that are later developed in the main text.
Reviewer point #3
Line 68, there are unexpected spaces, I guess.
Author response #3
We have noticed that the uploaded files suffered some changes during the submission process, which explain the mentioned mistake. We have mended the problem in the revised version of the manuscript.
Reviewer point #4
I think “RAs” might be Ras (line 73), and “The small GTPase Ras related nuclear protein (Ran)” is better.
Author response #4
The capital letters are for highlighting the letters that are included in the abbreviation. This is done throughout the text.
Reviewer point #5
Line 77, I think “or” should be “and”.
Author response #5
We have changed "or" to "and"
Reviewer point #6
I think that AP1 shoud be RanGAP1 (line 77).
Author response #6
Thank you for your comment. We have mended the mistake.
Reviewer point #7
Line 88. RanGEF has a specific name, RCC1.
Author response #7
Thank you, we have changed RanGEF to RCC1.
Reviewer point #8
Line 95, there are unexpected spaces, before TURC.
Author response #8
This depences once more of the changes that the text suffered upon submission. The problem is now mended.
Reviewer point #9
Figure 1 shows g-tubulin in the space between INM and ONM. Are there any references?
Author response #9
We have added the relevant references to line 193.
Reviewer point #10
Line 98, there are unexpected spaces, just before tubulin.
Author response #10
This depences once more of the changes that the text suffered upon submission. The problem is now mended.
Reviewer point #11
Line 124-140, the authors mentioned about nucleoli. Regarding to the subject of this review, I think that it had better to mention that the relationship between nucleoli and LINC complex.
Author response #11
Yes, we agree that this information should be added to the text. Please, see lines 226 and 227 in the revise version of the manuscript.
Reviewer point #12
The subtitle “Show me your nuclear architecture and I will tell you who you are” (line 141) is different from other subtitles.
Author response #12
Well, yes, but we think that the title describes well the content of the following chapter.
Thank you for your time and consideration.
Reviewer 3 Report
The manuscript submitted by Dr Corvaisier and Alvarado-Kristensson provides a very comprehensive overview about nuclear architecture and gene expression regulation by tubulin meshwork. The work is well structured, illustrated and referenced and authors are very well recognized in the field. Some attention could be brought to improve the last section of the Conclusion and perspectives to avoid some repetitions. Also, since the authors mention the relevance of TUBG genes for brain development and logically explain their involvement in brain tumor, it would be interesting to report, if any, more details about both the nuclear architecture and structural organization of the tubulin meshwork in that context or in tumors in general.
Some minor details should be considered before publication.
Overall editing comments:
- Many times symbols are missing, it is especially the case for γ.
- Check for consistency when introducing an abbreviation (i.e "microtubules" are sometimes referred as "microtubules" and sometimes as "MT", notably in section 4.6)
- Other minor changes to consider: l.46 an extra ")", l.54 "molds", l.67 "such as", l.148"assists", l.191 "Environmental, mechanical", l.280 "vary both in size and location", l.322 "electron microscopy", l.330 "to impair", l.388 "Tubulin in cancer"
- Title of section 3.4 could be more compelling.
- A reference could be added l.208 to support the notion of MAPs
Author Response
Reviewer point #1
Some attention could be brought to improve the last section of the Conclusion and perspectives to avoid some repetitions.
Author response #1
Thank you for your comments. We have removed the text that was repetitive in the conclusion and perspectives section. Please see lines 870 and 871.
Reviewer point #2
Also, since the authors mention the relevance of TUBG genes for brain development and logically explain their involvement in brain tumor, it would be interesting to report, if any, more details about both the nuclear architecture and structural organization of the tubulin meshwork in that context or in tumors in general.
Author response #2
This is a new field with few publications. To our knowledge, we have included the available publications in the field.
Reviewer point #3
Many times symbols are missing, it is especially the case for γ.
Author response #3
Yes, we have noticed that the uploaded files suffered some changes during the submission process. We have added back the missed information.
Reviewer point #4
Check for consistency when introducing an abbreviation (i.e "microtubules" are sometimes referred as "microtubules" and sometimes as "MT", notably in section 4.6)
Author response #4
Thank you for your comment. We have replaced microtubule with MT as suggested.
Reviewer point #5
Other minor changes to consider: l.46 an extra ")", l.54 "molds", l.67 "such as", l.148"assists", l.191 "Environmental, mechanical", l.280 "vary both in size and location", l.322 "electron microscopy", l.330 "to impair", l.388 "Tubulin in cancer"
Author response #5
Thank you for your comment. I have mended the mistakes.
Reviewer point #6
Title of section 3.4 could be more compelling.
Author response #6
As mentioned above, the uploaded files suffered some changes during the submission process that affected many of the titles. We have added back the missed information.
Reviewer point #7
A reference could be added l.208 to support the notion of MAPs
Author response #7
We have added to line 325 the requested reference.
Thank you for your time and consideration.